# Identification and quantification of cassava starch adulteration in different food starches by droplet digital PCR

Jia Chen[1], Yalun Zhang[2], Chen Chen[2], Yan Zhang[2], Wei Zhou[2]*, Yaxin Sang[1]*

**1** College of Food Science and Technology, Hebei Agricultural University, Baoding, Hebei, China, **2** Hebei Food Inspection and Research Institute, Hebei Food Safety Key Laboratory, Shijiazhuang, China

* sangyaxin@sina.com (YS); zhouwei0311@163.com (WZ)

**Data Availability Statement:** All relevant data are within the paper and its Supporting Information files.

## Abstract

We report a rapid and accurate quantitative detection method using droplet digital PCR (ddPCR) technology to identify cassava adulteration in starch products. The ddPCR analysis showed that the weight of cassava (M) and cassava-extracted DNA content had a significant linear relationship—the correlation coefficient was $R^2 = 0.995$, and the maximum coefficient of variation of replicates was 7.48%. The DNA content and DNA copy number ($C$) measured by ddPCR also had a linear relationship with $R^2 = 0.992$; the maximum coefficient of variation of replicates was 8.85%. The range of cassava ddPCR DNA content was 25 ng/µL, and the formula $M = (C + 32.409)/350.579$ was obtained by converting DNA content into the median signal. The accuracy and application potential of the method were verified using the constructed adulteration model.

## 1. Introduction

Starch is a staple and major source of calories and is often used in modern food industries. The most common types of starch in China are potato, sweet potato, cassava, corn, and wheat as defined by the China National Standard for Starch Products GB 2713–2015. The price difference is due to the availability of raw materials and the cost of production. These price differences can lead to starch adulteration [1]. Adulteration not only causes economic loss to customers but can also lead to risks of food allergy. Cassava starch is the main material in adulteration of more expensive starches, and sensitive detection techniques are thus needed to detect and deter adulteration [2]. The detection of starch mainly includes sensory tests as well as physical and chemical tests. However, these test methods are time-consuming and labor-intensive and cannot measure the extent of adulteration. Accurately identifying the degree of adulteration is difficult. Establishing a precise, rapid, and effective quantitative analysis method is thus very important.

Molecular biology can help identify adulterants via multiplex PCR, fluorescent PCR, digital PCR (ddPCR), and other PCR technologies [3]. These tools are Sensitive, fast, and useful in food science [4]; they have gradually replaced colorimetric detection methods, but these applications cannot accurately quantify adulterants in processed starches [5–6].

At the end of the 20th century, Brunetto et al. [7] proposed the concept of digital ddPCR, which distributes sample DNA evenly into a large number of reaction units and then independently performs PCR amplification on each reaction unit. The ddPCR can obtain a DNA copy

**Funding:** This work was supported by South China Agricultural University [grant number 2017YFC1601700].

**Competing interests:** The authors have declared that no competing interests exist.

number without reference to standard curve or control gene [8–10]. ddPCR offers good sensitivity, high precision, and absolute quantification. It has been analyzed in terms of copy-number variation [11,12], transgenic properties [13,14], single nucleotide polymorphisms [15], gene expression analysis [16], and microbial detection [17,18]. This technology has important application prospects because of its quantitative nature.

Here, the relationship between cassava weight and the extraction efficiency of cassava starch (i.e., tapioca) DNA was first studied. Specific primers for a cassava gene were then designed, and ddPCR technology was used to quantify the DNA and establish the relationship between extracted DNA concentration and amplified DNA copy number. This established a formula for calculating cassava weight from the copy number. A adulteration model of sweet potato and cassava was constructed to explore the applicability of this method via 50 different commercially available starch verification methods. Finally, a rapid, accurate, and quantitative detection method for cassava adulterants was constructed to complement the quantitative testing technology of cassava in starch.

## 2. Methods

### 2.1. Test materials

Sweet potato starch and tapioca were obtained from a food-processing plant (Convenience Farmer's Comprehensive Market, Nanchang Street, West Bridge District, Shijiazhuang, Hebei, China). In addition, 2 × ddPCR supermix for probes, droplet generation oil, and droplet reader oil were purchased from Bio-Rad. Primers were synthesized by Shanghai Bioengineering Co., Ltd. A deep processing food DNA extraction kit (Tiangen Company) as well as analytically pure isopropanol and anhydrous ethanol were purchased from Beijing Luqiao Company.

### 2.2. Experimental methods

**2.2.1. DNA extraction.** DNA extraction starch product was performed according to the manufacturer's instructions. Here, 100 mg of the sample was added to 500 μL of buffer GMO1 and 20 μL of proteinase K (20 mg/mL); this was vortexed for 1 min. The solution was then incubated at 56˚C for 1 h and oscillated every 15 min during the incubation. Next, 200 μL of buffer GMO2 was added and mixed well and vortexed for 1 min with 10 min of subsequent incubation at room temperature. The solution was then centrifuged at 12,000 rpm for 5 min, and the supernatant was aspirated into another centrifuge tube. Next, 0.7 mL of isopropanol was added to the supernatant and mixed well. The solution was then centrifuged at 12,000 rpm for 3 min to remove the supernatant and retain the pellet. We then added 700 μL of 70% ethanol, vortexed for 5 s, centrifuged at 12,000 rpm for 2 min on a centrifuge, and removed the supernatant. This was repeated a second time. We then opened the lid in the biosafety cabinet for 20 min to thoroughly dry the residual ethanol. Next, 20–50 μL of elution buffer TE was added and vortexed for 1 min to obtain a DNA solution. The quantity and purity of DNA were determined via a nucleic acid analyzer (NanoDrop 2000 by Thermo).

**2.2.2. Reaction primer design.** Primers were designed using the primer design software DNAman and Primer Premier 5.0. The cassava-specific primer sequence was designed based on the intergenic spacer of chloroplast *trnL-trnF* sequence—this is one of the most frequently used molecular markers of plants [19] (GenBank: EU518905.1) [20] (Table 1).

**2.2.3. ddPCR reaction system.** Here, a 20 μL amplification system was used that contained 10 μL 2 × ddPCR supermix; the forward primer concentration was 10 μmol/L (1.2 μL used), and the reverse primer concentration was 10 μmol/L (1.2 μL used). The concentration was 10 μmol/L (0.4 μL used) with 4.0 μL of DNA template. The balance was ddH$_2$O. Sterile ddH$_2$O was used as a blank control.

**Table 1. Cassava gene primer sequence.**

|  | F-Primer 5′-3′ | R-Primer 5′-3′ | Probe |
|---|---|---|---|
| Cassava | GGGGGATAGGTGCAGAGACT | AAAAATACGGATTTGGGCCCCT | FAM- TGGAGTTGACTGCGTTGCATTAGT–TAMRA |

**2.2.4. Main operating procedures for ddPCR reactions.** The fully mixed PCR reaction system was transferred to a droplet-generating card (Bio-Rad); 70 μL of the droplet-generation oil was added to the droplet-generating card, and the droplets were carded into a droplet generator (Bio-Rad) for reaction. The resulting droplets were then transferred to 96 wells of ddPCR, and the 96-well plates were sealed to prepare for the PCR reaction.

The program for the reaction denaturation was as follows: 95˚C, 10 min; 94˚C denaturation, 1 min; 56˚C annealing, 45 s; 40 cycles; 98˚C, 10 min; and 4˚C for temporary storage.

After the ddPCR reaction was completed, the 96-well plate was placed in the QX200 Droplet Reader (Bio-Rad), and the sample information was sequentially input. At the beginning of the test, the instrument automatically recognizes the droplets of each sample in sequence, and the droplets were sequentially passed through two-color detection via the droplet-reading oil. The positive and negative results were determined based on the intensity of the fluorescent signal emitted by the droplets, and the number of positive and negative droplets per sample was recorded. The results were calculated using Quantasoft software after signal acquisition was completed.

**2.2.5. Specific detection of cassava DNA by ddPCR.** The primers specific to cassava were used to digitally amplify the genomic DNA of starch from sweet potato, cassava, potato, corn and sesame, walnut, soybean, hazelnut, beef, mutton; sterile ddH$_2$O was used as a blank control to determine the specificity of the primer. The experimental system and operating procedures are shown above 2.2.3 and 2.2.4.

**2.2.6. Determination of conversion formula between cassava weight and the copy number of ddPCR.**

*2.2.6.1. Cassava weight and extracted DNA concentration.* First, 5–100 mg of cassava samples was weighed for DNA extraction and three replicates were performed to ensure the repeatability of the experiment. They were then evaluated with Nanodrop 2000. We then established the relationship between the weight of the sample and the amount of DNA finally extracted.

*2.2.6.2. Establishment of the relationship between the copy number from ddPCR and the concentration of cassava DNA.* We first evaluated the relationship between copy number and DNA concentration as well as the proportion and weight of adulterated substances. The extracted DNA was diluted to a gradient of 1, 5, 10, 15, 20, and 25 ng/μL. This was then amplified and detected using ddPCR technology. Sterile ddH$_2$O was used as a blank control. The experimental system and operating procedures are shown above in 2.2.3 and 2.2.4.

**2.2.7. Construction of an adulteration model of sweet potato and cassava.** An artificially constructed adulteration model of sweet potato and cassava was used to simulate the adulteration of other starch products with tapioca. This was used to evaluate ddPCR technology as a tool to identify cassava adulteration. Sweet potato starch and tapioca were mixed in different ratios for DNA extraction (Table 2). Then, DNA was extracted from 10 mg of the mixed starch samples, and 4 μL of extracted DNA was used in ddPCR. The correlation between the weight and DNA copy number of tapioca was calculated using Origin 8.6 to

**Table 2. Artificially simulate the adulteration of cassava starch in sweet potato starch.**

| Sample | A:B[a] | A:B | A:B | A:B | A:B | A:B | A:B | A:B | A:B |
|---|---|---|---|---|---|---|---|---|---|
| Mass (mg) | 90:10 | 80:20 | 70:30 | 60:40 | 50:50 | 40:60 | 30:70 | 20:80 | 10:90 |

[a] A is cassava, and B is sweet potato.

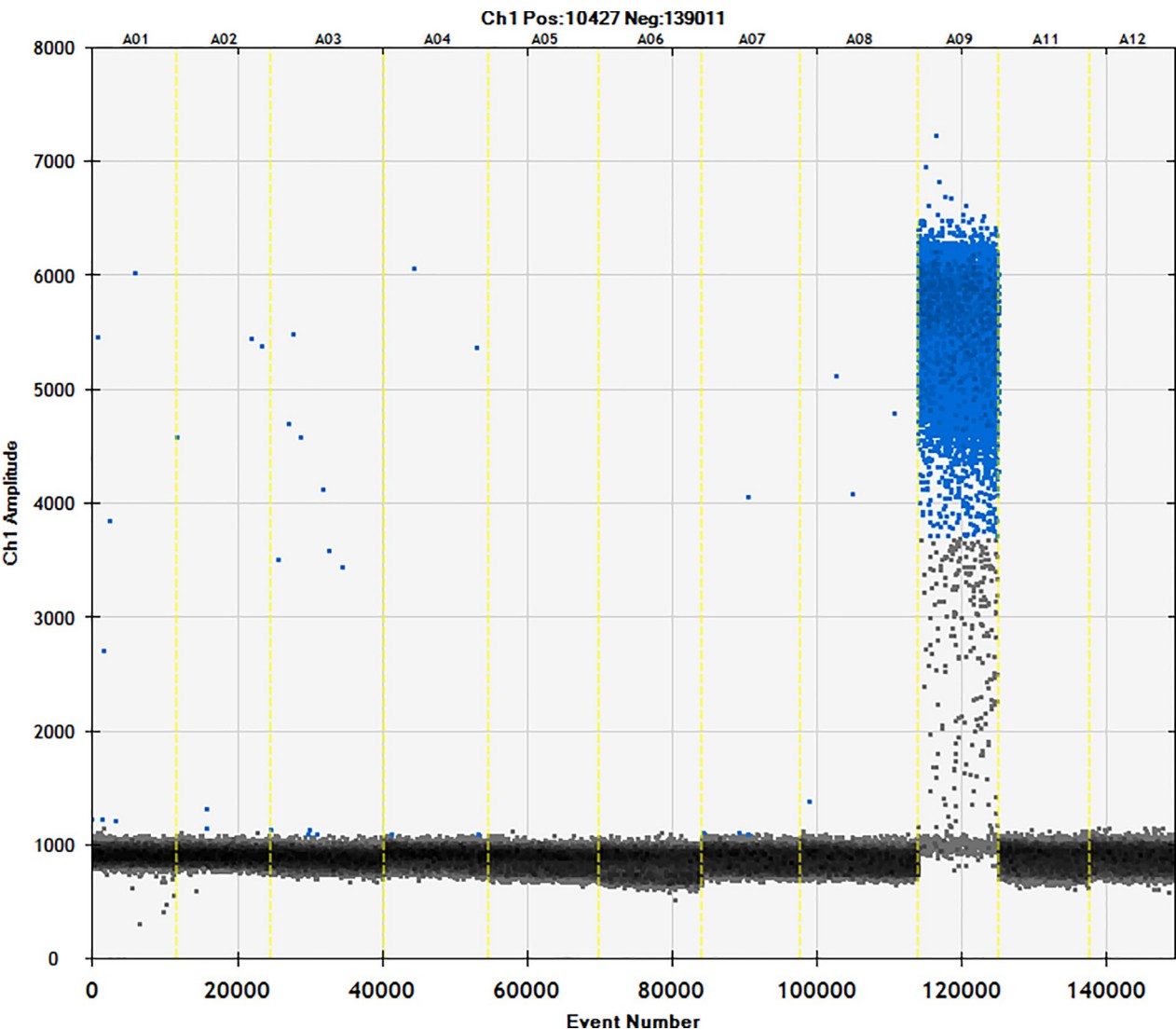

**Fig 1. Validation of the specificity of cassava primers.** The specificity of cassava primers were tested using the following samples: 1, beef; 2, lamb; 3, hazelnut; 4, soybean; 5, walnut; 6, sesame; 7, corn starch; 8, potato starch; 9, cassava starch; 10, sweet potato starch; and 11, ddH$_2$O.

obtain a formula for calculating the cassava weight from the DNA copy number. The measured values of cassava weight based on this formula were compared with the actual values of cassava weight to evaluate the practical utility of this method.

**2.2.8. Commercial sample inspection.** To further verify the practical utility of this method, 50 different brands of starch were purchased from large supermarkets and farmers' markets including 30 products of sweet potato starch, 12 products of potato starch, and 8 products of corn starch. Each sample was analyzed three times by ddPCR as described above.

## 3. Results

### 3.1. Specific detection of ddPCR

The cassava-specific primers were used with sweet potato, cassava, potato, corn, sesame, walnut, soybean, hazelnut, beef, and mutton. Primers were specific with no cross-reactivity with other starches tested (Fig 1). This has value in subsequent detection.

**Table 3. Cassava DNA extraction results.**

| Sample name | Mass (mg) | DNA content (ng/μL) | | | Average value (ng/μL) | Coefficient of variation (%) |
|---|---|---|---|---|---|---|
| | | **#1** | **#2** | **#3** | | |
| Cassava | 5.0 | 6.2 | 6.1 | 5.7 | 6 | 4.41 |
| | 10.0 | 13.1 | 11.4 | 12.5 | 12.3 | 6.99 |
| | 20.0 | 26 | 23.5 | 25.4 | 25 | 5.23 |
| | 30.0 | 44.3 | 40.7 | 38.8 | 41.3 | 6.77 |
| | 40.0 | 53.7 | 56.9 | 54.5 | 55 | 3.03 |
| | 50.0 | 68.4 | 63.3 | 68.1 | 66.6 | 4.30 |
| | 60.0 | 80.8 | 82.3 | 71.5 | 78.2 | 7.48 |
| | 70.0 | 93.3 | 88.2 | 92.8 | 91.4 | 3.07 |
| | 80.0 | 110.8 | 100.5 | 99.8 | 103.7 | 5.94 |
| | 90.0 | 122.1 | 114.2 | 107.5 | 114.6 | 6.38 |
| | 100.0 | 139.5 | 132.8 | 144.8 | 139 | 4.33 |

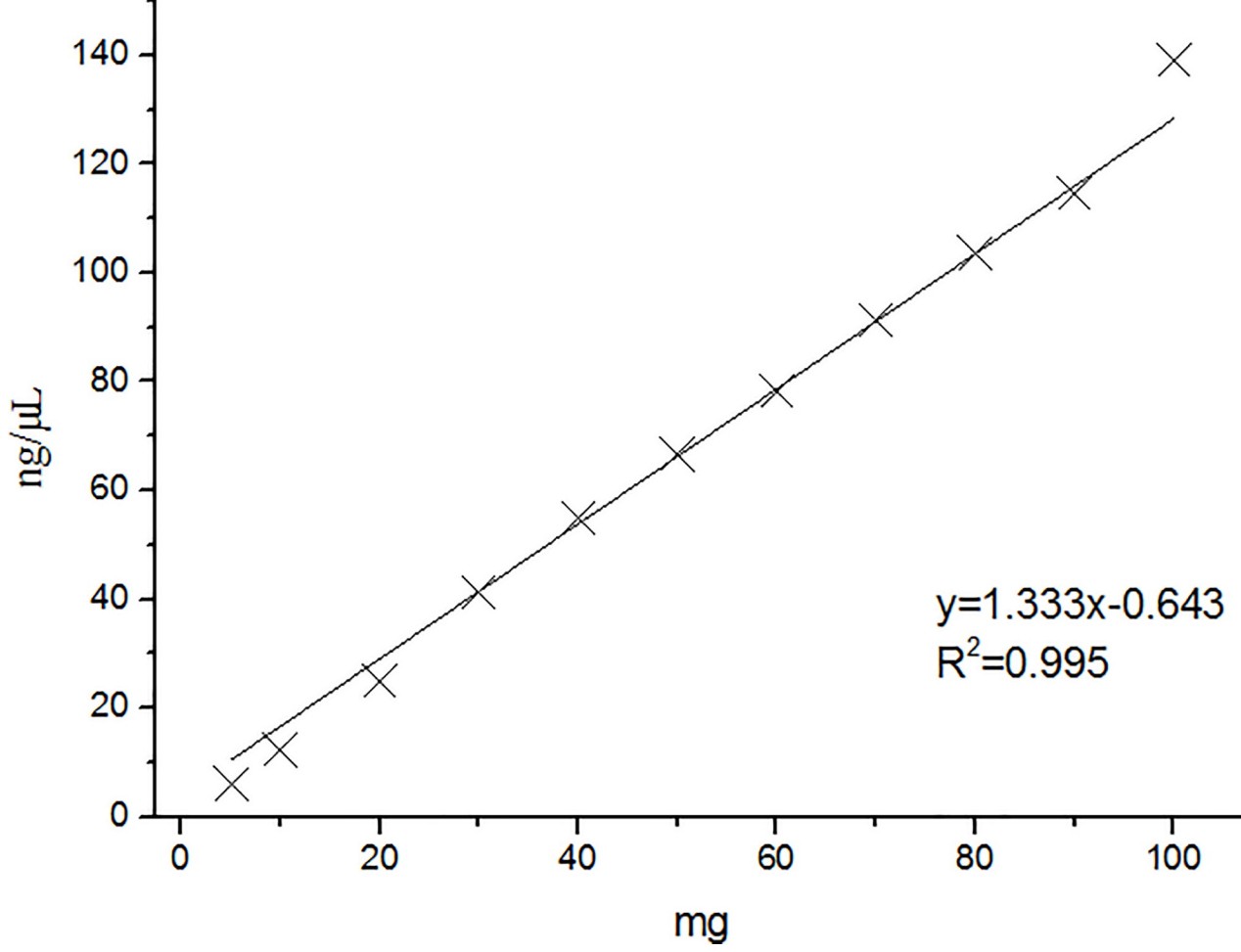

$$y=1.333x-0.643$$
$$R^2=0.995$$

**Fig 2. Correlation between tapioca dry weight and extracted DNA.**

**Table 4. Cassava copy number under gradient DNA content.**

| Sample name | DNA content (ng/µL) | Copy number (copies/µL) | | | Average value (copies/µL) | Coefficient of variation (%) |
|---|---|---|---|---|---|---|
| | | #1 | #2 | #3 | | |
| Cassava | 1 | 289 | 326 | 294 | 303 | 6.63 |
| | 5 | 1739 | 1797 | 1566 | 1700.7 | 7.07 |
| | 10 | 2510 | 2896 | 2970 | 2792.3 | 8.85 |
| | 15 | 4147 | 3567 | 3936 | 3883.3 | 7.56 |
| | 20 | 4939 | 5649 | 4973 | 5186.7 | 7.72 |
| | 25 | 7533 | 6327 | 6970 | 6943.3 | 8.69 |

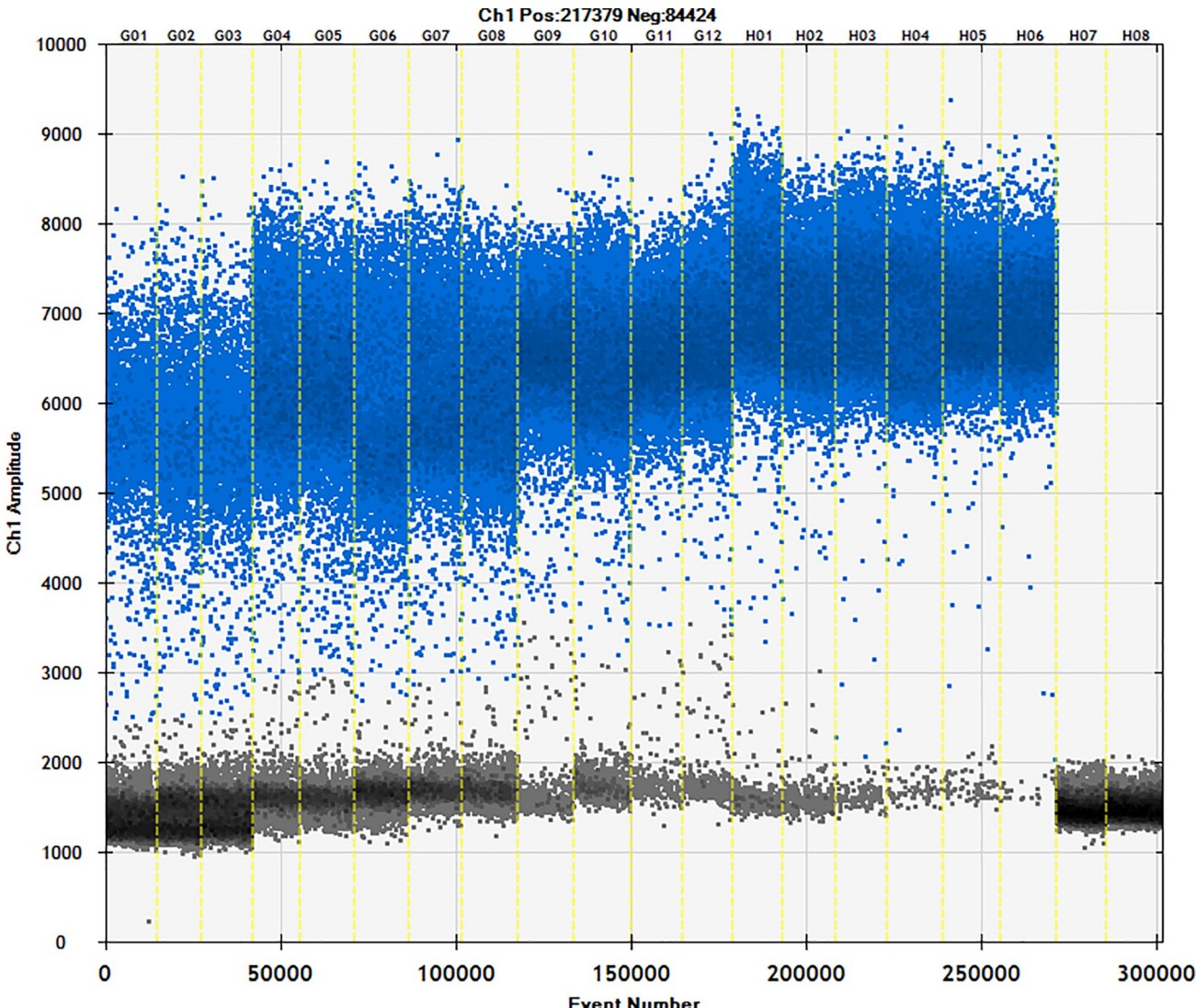

**Fig 3. Correlation between the content and copy number of cassava starch DNA.** DNA concentration was measured by NanoDrop 2000 and DNA copy number was determined by ddPCR.

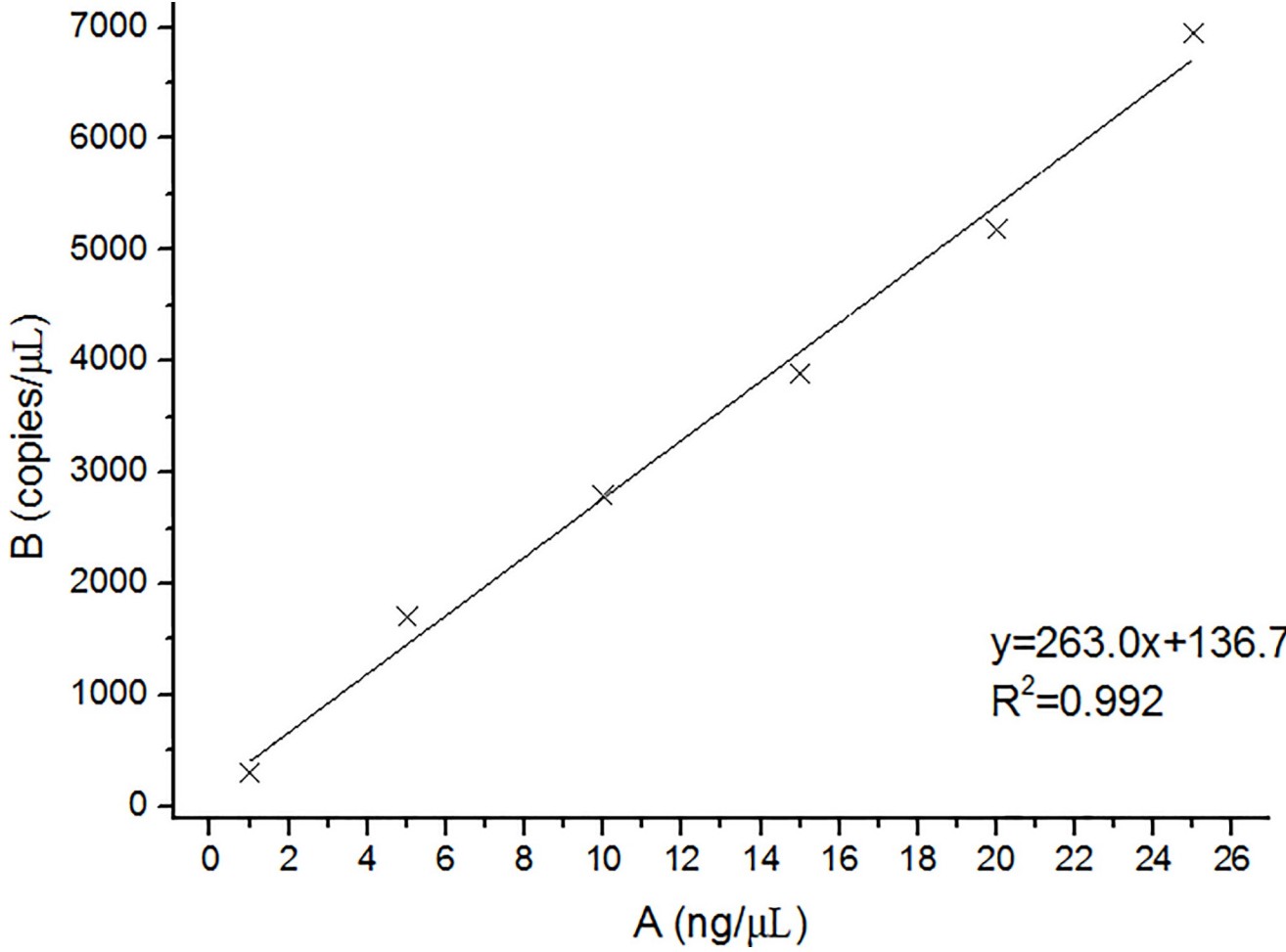

**Fig 4. Relationship between cassava DNA content and copy number.**

## 3.2. Determination of the conversion formula between the weight of cassava and the copy number of ddPCR

**3.2.1. Cassava weight and extracted DNA concentration.** Eleven weight groups ranging from 5.0 to 100.0 mg of tapioca (Table 3). The maximum coefficient of variation of replicates was 7.48%, which is much lower than the specified requirement coefficient of variation of 15%. This suggests that the data were stable and reliable. The average of the three replicates of the extracted DNA results was linearly fitted to the cassava weight and found to be linear (Fig 2); the correlation coefficient $R^2$ was 0.995.

**Table 5. Establishment of cassava dose response curve.**

| Linear curve formula | $R^2$ |
|---|---|
| $C_{DNA} = 1.333M - 0.643$ | 0.995 |
| $C = 263.0\ C_{DNA} + 136.7$ | 0.992 |
| $M = (C + 32.409)/350.579$ | |

$C_{DNA}$ = DNA concentration, $C$ = Copy numbers, M = cassava mass

1 2　3 4　5 6　7 8　910111213141516 1718 19 20 21 2223 24 25 26

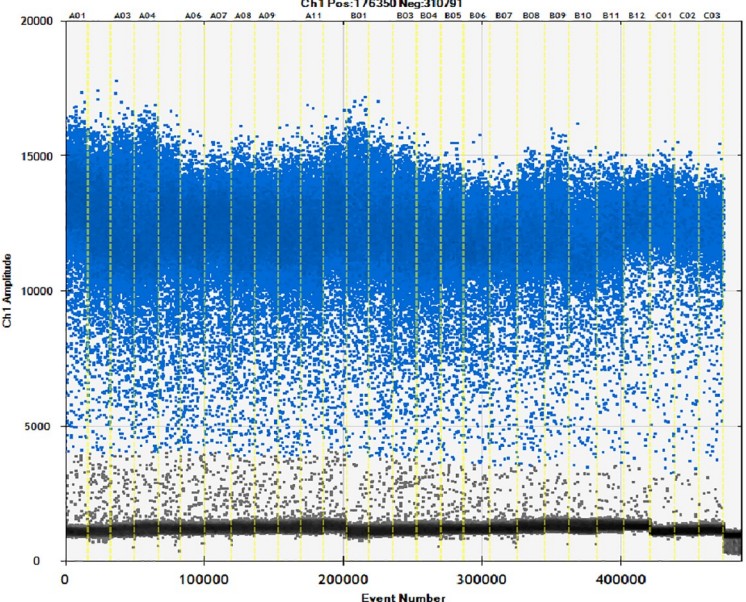

**Fig 5. Copy number of cassava and sweet potato ratio.** Channels 1–3: starch mixture containing 90% cassava starch; channels 4–6: starch mixture containing 80% cassava starch; channels 7–9: starch mixture containing 70% cassava starch; channels 10–12: starch mixture containing 60% cassava starch; channels 13–15: starch mixture containing 50% cassava starch; channels 16–18: starch mixture containing 40% cassava starch; channels 19–21: starch mixture containing 30% cassava starch; channels 22–24: starch mixture containing 20% cassava starch; channels 25–27: starch mixture containing 10% cassava starch; and channel 28: sterile double-distilled water.

**3.2.2. Detection of the relationship between the copy number of ddPCR and the concentration of cassava DNA.** The extracted DNA was diluted to a gradient of 1, 5, 10, 15, 20, and 25 ng/μL; there were three replicates for each concentration, and 4 μL of DNA was used for ddPCR. The results are shown in Table 4. The copy number of cassava increased with increased DNA content. There was a significant linear relationship. The coefficient of variation was 8.85%, and this was far below the coefficient of variation required by the regulations. The average copy number and DNA content of the three replicates were linearly fitted (Figs 3 and 4) with a correlation coefficient $R^2$ of 0.992.

**Table 6. Analysis results of cassava with known adulterants.**

| Number | Cassava mass (mg)[a] | DNA Copy number (copies/μL) | | | Average value (copies/μL) | Coefficient of variation (%) | Measured cassava mass (mg) | Relative error (%) |
|---|---|---|---|---|---|---|---|---|
| 1 | 10.0 | 320.9 | 347.2 | 309.3 | 325.8 | 5.96 | 10.22 | 2.2 |
| 2 | 20.0 | 701 | 704 | 668 | 691 | 2.89 | 20.63 | 3.15 |
| 3 | 30.0 | 878 | 961 | 972 | 937 | 5.48 | 27.65 | −7.83 |
| 4 | 40.0 | 1495 | 1635 | 1409 | 1513 | 7.54 | 44.08 | 10.2 |
| 5 | 50.0 | 1723 | 1715 | 1581 | 1673 | 4.77 | 48.65 | −2.7 |
| 6 | 60.0 | 2309 | 2087 | 2255 | 2217 | 5.22 | 64.16 | 6.93 |
| 7 | 70.0 | 2439 | 2504 | 2248 | 2397 | 5.55 | 69.3 | −1 |
| 8 | 80.0 | 2838 | 2629 | 2822 | 2763 | 4.21 | 79.74 | −0.32 |
| 9 | 90.0 | 3173 | 3318 | 3490 | 3327 | 4.77 | 95.82 | 6.47 |

[a] The total mass-of the cassava and sweet potato starch mixture was 100 mg.

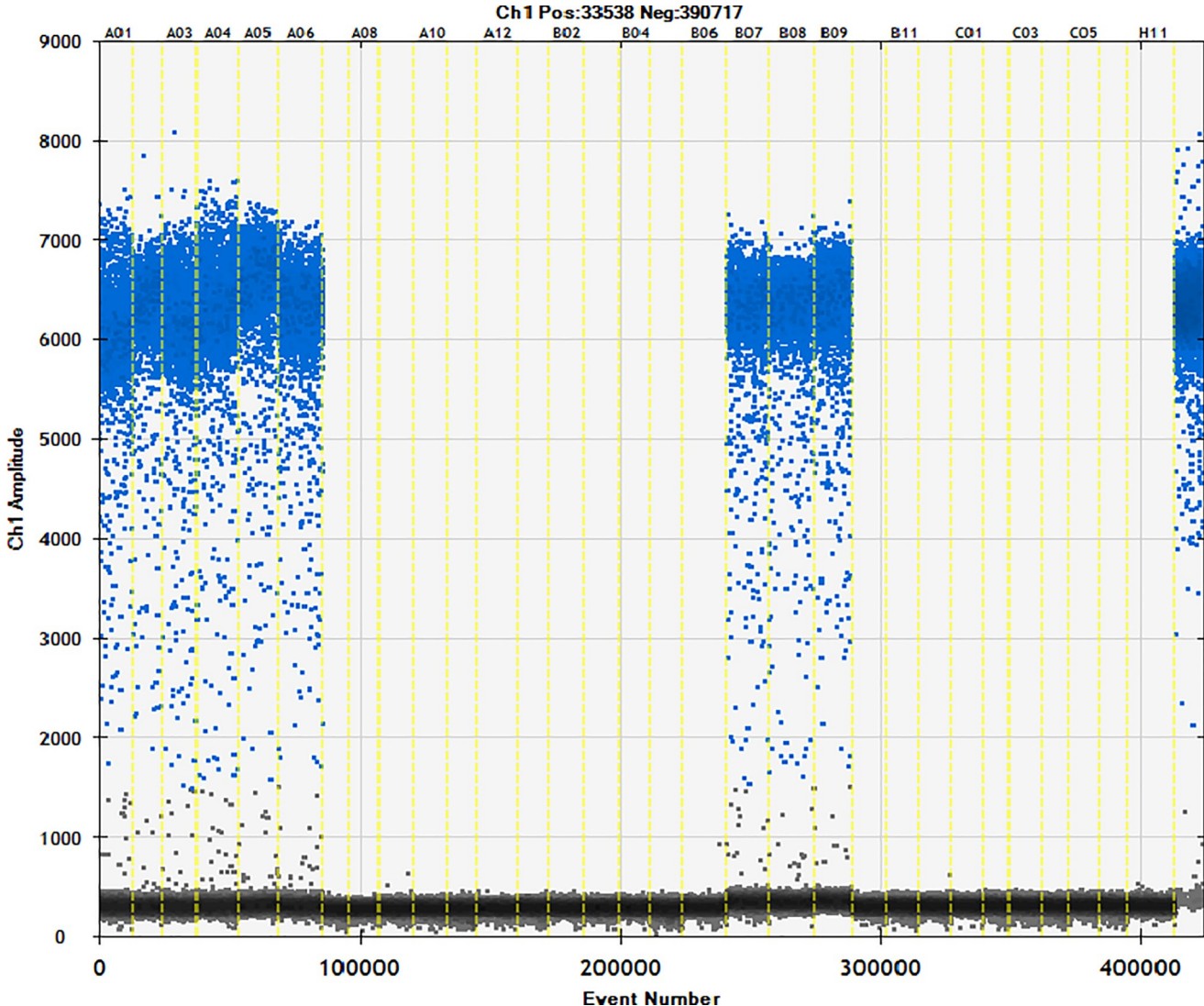

**Fig 6. Actual test of commercially available samples.** Channel 1–3: sample1; channel 4–6: sample2; channel 7–9: sample3; channel 10–12: sample4; channel 13–15: sample5; channel 16–18: sample6; channel 19–21: sample7; channel 22–24: sample8; channel 25–27: sample9; channel 28–30: sample10; channel31 negative; channel32 positive.

**3.2.3. Determination of the relationship between the weight of cassava and the copy number of ddPCR.** There was a significant linear relationship between the weight of cassava (M) and the content of extracted cassava DNA. And there was a certain linear relationship between the DNA content and DNA copy number (C) of cassava. Using the cassava DNA content as the intermediate conversion value, the formula is obtained for the cassava quality and the cassava DNA copy number (Table 5). The formula is $M = (C + 32.409)/350.579$ where M is the cassava mass (mg), and C is the amplified DNA copy number (copies/μL).

## 3.3. Method validation—Construction of sweet potato and cassava adulteration model

The cassava and sweet potato starch were mixed at ratios of 1:9 to 9:1 to a total of 100 mg. DNA was extracted from 10 mg of mixed starch samples, and 4 μL was taken for ddPCR. The

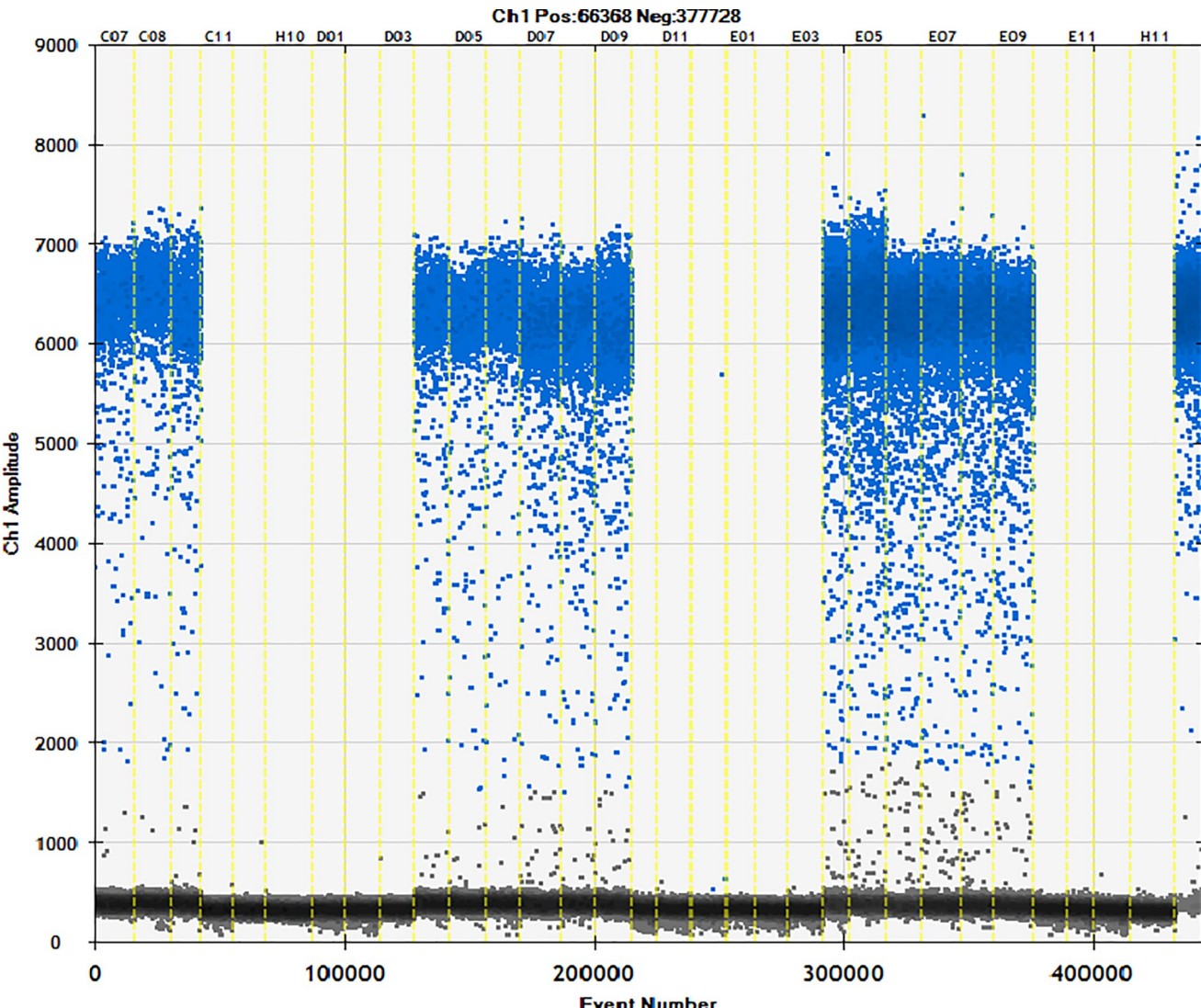

**Fig 7. Actual test of commercially available samples.** Channel 1–3: sample11; channel 4–6: sample12; channel 7–9: sample13; channel 10–12: sample14; channel 13–15: sample15; channel 16–18: sample16; channel 19–21: sample17; channel 22–24: sample18; channel 25–27: sample19; channel 28–30: sample20; channel31 negative; channel32 positive.

amplification results are shown in Fig 5. The results (Table 6) show that the coefficient of variation between copy numbers was 7.54%, which is much lower than the coefficient of variation required by the regulations. The weight of the cassava in the combined sample was consistent with the actual weight, and the maximum relative error value was 10.2%. This was also within the specified error range. The accuracy and precision of the ddPCR method established here were thus verified using the sweet potato and cassava adulteration model. This suggests that the method can detect cassava in commercial starch products.

### 3.4. Detection of commercially available samples

Fifty different brands of starch were studied using the ddPCR method (Figs 6–10). The total starch weight used was 10 mg, and 4 µL of extracted DNA was taken for ddPCR. The average value of three replicate experiments was calculated (Table 7). The highest ratio of cassava

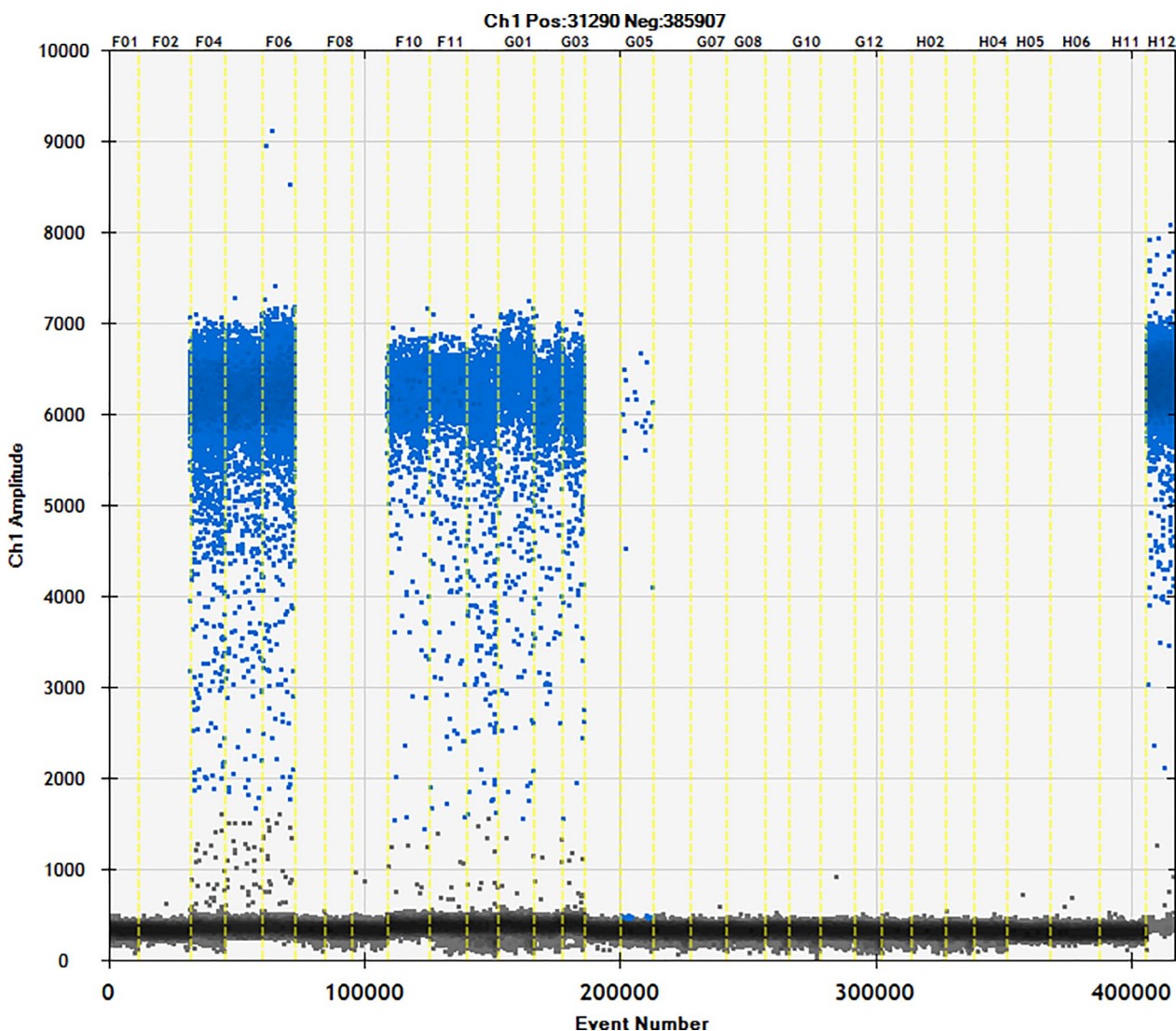

**Fig 8. Actual test of commercially available samples.** Channel 1–3: sample21; channel 4–6: sample22; channel 7–9: sample23; channel 10–12: sample24; channel 13–15: sample25; channel 16–18: sample26; channel 19–21: sample27; channel 22–24: sample28; channel 25–27: sample29; channel 28–30: sample30; channel31 negative; channel32 positive.

adulteration in sweet potato starch was 37.38%, and 11 of the 30 sweet potato starch products had cassava adulteration. The highest measured cassava adulteration in potato starch was 9.65%, and 11 of the 30 sweet potato starch products had cassava adulteration. The highest ratio of cassava adulteration in corn starch was 10.37%, and there were 2 out of 8 samples with cassava adulteration. These results show that cassava adulteration can be quantitatively identified.

## 4. Discussion

ddPCR was used to accurately and quantitatively detect cassava-derived components in starch. A linear relationship among cassava weight, DNA concentration, and amplified DNA copy number was discovered. The calculation formula of weight and amplified DNA copy number

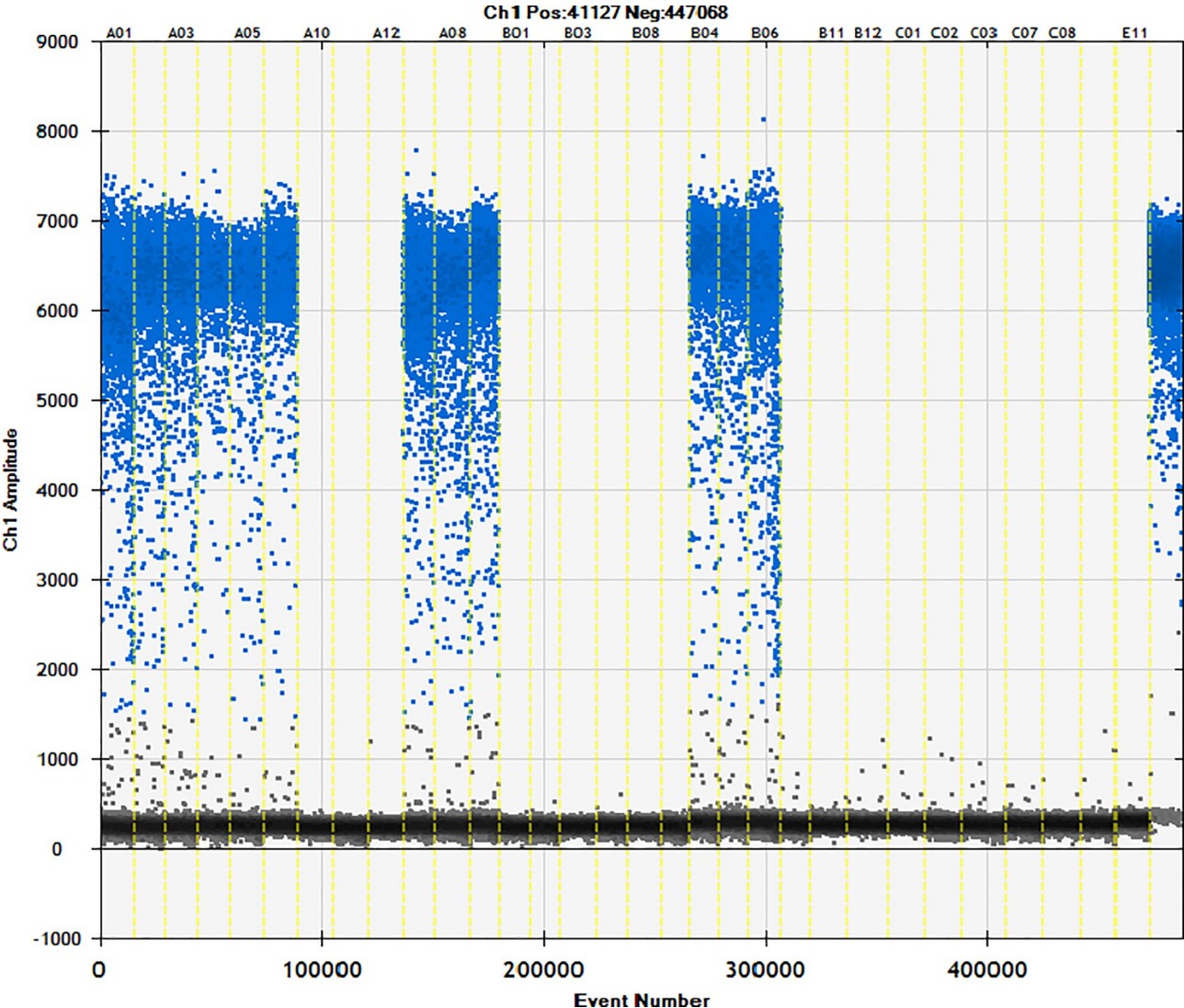

**Fig 9. Actual test of commercially available samples.** Channel 1–3: sample31; channel 4–6: sample32; channel 7–9: sample23; channel 10–12: sample24; channel 13–15: sample25; channel 16–18: sample26; channel 19–21: sample27; channel 22–24: sample28; channel 25–27: sample29; channel 28–30: sample30; channel31 negative; channel32 positive.

can quickly report the cassava content for quantitative detection of adulterants in commercial starch products.

This study confirms the market applicability and accuracy of the method via a mixture of sweet potato and cassava starch of different ratios. The ddPCR amplification results are largely consistent with the actual weight. The maximum relative error value is 10.2%, which is within the specified error range Furthermore, statistical analysis showed that the difference between the replicate measurements is small (low coefficient of variation). These data show that this approach is reliable and can measure cassava adulteration.

In order to verify the application prospect of this study, 50 starch products of different brands were tested and analyzed. The highest weight of cassava adulteration in sweet potato starch was 37.38%, and 11 out of 30 samples had cassava adulteration. The highest ratio of cassava adulteration in potato starch was 9.65%. There were 5 samples in 12 samples with cassava

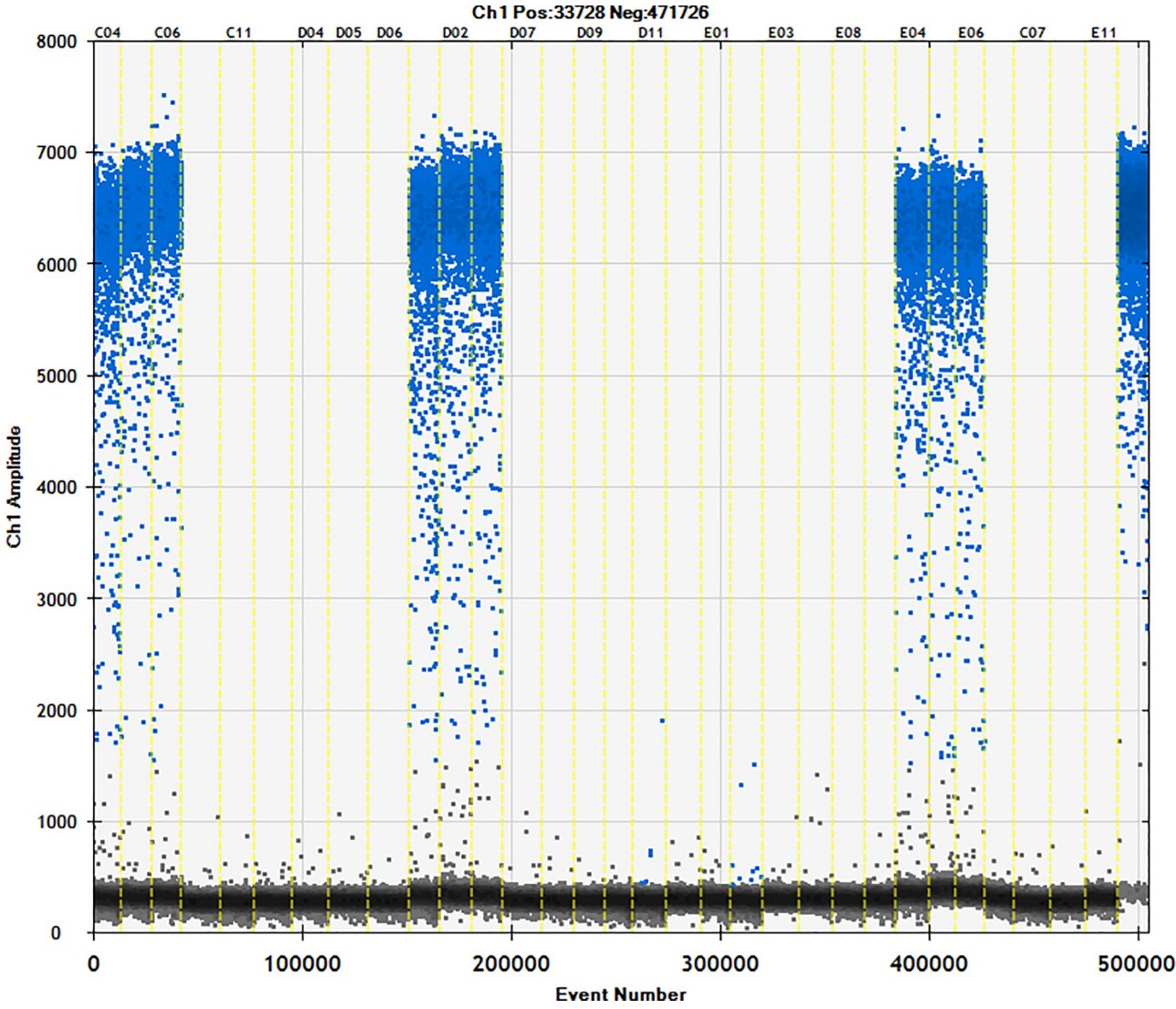

**Fig 10. Actual test of commercially available samples.** Channel 1–3: sample41; channel 4–6: sample42; channel 7–9: sample43; channel 10–12: sample44; channel 13–15: sample45; channel 16–18: sample46; channel 19–21: sample47; channel 22–24: sample48; channel 25–27: sample49; channel 28–30: sample50; channel31 negative; channel32 positive.

adulteration. The highest measured cassava adulteration in corn starch was 10.37%; this was seen in 2 of 8 samples. The results of this series of tests indicate that there are different degrees of adulteration in commercially available starch products indicating the necessity to develop efficient detection approaches. Our method can accurately and quantitatively measure the degree of adulteration of commercially available starch. These findings may help distinguish deliberate adulteration from contamination. For example, some weight ratios of up to 10% and 30% adulteration must be deliberately adulterated, while some 3% and 4% may be due to contamination during production processes.

The method can quantitatively determine the degree of adulteration in commercially available starch. It can also work with a wide range of adulterants. Thus, ddPCR technology can discriminate between intentional fraud and unintended contamination. The method can also be applied to other types of starch testing, and this quantitative testing system is a valuable tool

**Table 7. Analysis of commercially available samples.**

| Sample name | Number | Copy number (copies/μL) | | | Average value (copies/μL) | Adulteration mass ratio (%) |
|---|---|---|---|---|---|---|
| | | #1 | #2 | #3 | | |
| Sweet potato starch | 1 | 331 | 348 | 305 | 328 | 10.28 |
| | 2 | 237.1 | 233.5 | 262 | 244.2 | 7.89 |
| | 3 | 0 | 0 | 0 | 0 | 0 |
| | 4 | 0 | 0 | 0 | 0 | 0 |
| | 5 | 0 | 0 | 0 | 0 | 0 |
| | 6 | 0 | 0 | 0 | 0 | 0 |
| | 7 | 186 | 193.1 | 190.6 | 189.9 | 6.34 |
| | 8 | 0 | 0 | 0 | 0 | 0 |
| | 9 | 0 | 0 | 0 | 0 | 0 |
| | 10 | 0 | 0 | 0 | 0 | 0 |
| | 11 | 129 | 136 | 131 | 132 | 4.69 |
| | 12 | 0 | 0 | 0 | 0 | 0 |
| | 13 | 0 | 0 | 0 | 0 | 0 |
| | 14 | 132.3 | 133.1 | 139 | 134.8 | 4.77 |
| | 15 | 383 | 434 | 407 | 408 | 12.57 |
| | 16 | 0 | 0 | 0 | 0 | 0 |
| | 17 | 0 | 0 | 0 | 0 | 0 |
| | 18 | 1263 | 1336 | 1235 | 1278 | 37.38 |
| | 19 | 449 | 412 | 456 | 439 | 13.45 |
| | 20 | 0 | 0 | 0 | 0 | 0 |
| | 21 | 0 | 0 | 0 | 0 | 0 |
| | 22 | 468 | 442 | 473 | 461 | 14.07 |
| | 23 | 0 | 0 | 0 | 0 | 0 |
| | 24 | 109 | 96 | 107 | 104 | 3.89 |
| | 25 | 439 | 445 | 454 | 446 | 13.65 |
| | 26 | 0 | 0 | 0 | 0 | 0 |
| | 27 | 0 | 0 | 0 | 0 | 0 |
| | 28 | 0 | 0 | 0 | 0 | 0 |
| | 29 | 0 | 0 | 0 | 0 | 0 |
| | 30 | 0 | 0 | 0 | 0 | 0 |
| Potato starch | 1 | 302 | 290 | 326 | 306 | 9.65 |
| | 2 | 152 | 161 | 152 | 155 | 5.35 |
| | 3 | 0 | 0 | 0 | 0 | 0 |
| | 4 | 307 | 273 | 269 | 283 | 9.0 |
| | 5 | 0 | 0 | 0 | 0 | 0 |
| | 6 | 0 | 0 | 0 | 0 | 0 |
| | 7 | 218 | 191 | 188 | 199 | 6.60 |
| | 8 | 0 | 0 | 0 | 0 | 0 |
| | 9 | 0 | 0 | 0 | 0 | 0 |
| | 10 | 0 | 0 | 0 | 0 | 0 |
| | 11 | 139.5 | 125.1 | 127.2 | 130.6 | 4.65 |
| | 12 | 0 | 0 | 0 | 0 | 0 |

(*Continued*)

**Table 7.** (Continued)

| Sample name | Number | Copy number (copies/μL) | | | Average value (copies/μL) | Adulteration mass ratio (%) |
|---|---|---|---|---|---|---|
| | | #1 | #2 | #3 | | |
| Corn starch | 1 | 0 | 0 | 0 | 0 | 0 |
| | 2 | 331 | 325 | 337 | 331 | 10.37 |
| | 3 | 0 | 0 | 0 | 0 | 0 |
| | 4 | 0 | 0 | 0 | 0 | 0 |
| | 5 | 0 | 0 | 0 | 0 | 0 |
| | 6 | 0 | 0 | 0 | 0 | 0 |
| | 7 | 253 | 231 | 227 | 237 | 7.68 |
| | 8 | 0 | 0 | 0 | 0 | 0 |

for surveillance of quality control, maintenance of regulatory standards and consumer advocacy.

## Acknowledgments

This thesis was completed under the guidance of Prof. Yaxin Sang. His serious scientific attitude, rigorous academic spirit, and good work style inspired me. I also would like to thank Wei Zhou and Yalun Zhang for their help.

## Author Contributions

**Conceptualization:** Wei Zhou.

**Writing – original draft:** Jia Chen.

**Writing – review & editing:** Yalun Zhang, Chen Chen, Yan Zhang, Wei Zhou, Yaxin Sang.

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
