## [Decision Letter · Decision Letter 0]

12 Aug 2019

PONE-D-19-17769

A quantitative study of the adulteration of cassava components in starch products by droplet digital PCR

PLOS ONE

Dear Dr. Yaxin Sang

Thank you for submitting your manuscript to PLOS ONE. After careful consideration, we feel that it has merit but does not fully meet PLOS ONE’s publication criteria as it currently stands. Therefore, we invite you to submit a revised version of the manuscript that addresses the points raised during the review process.

We would appreciate receiving your revised manuscript by Sep 26 2019 11:59PM. To enhance the reproducibility of your results, we recommend that if applicable you deposit your laboratory protocols in protocols.io, where a protocol can be assigned its own identifier (DOI) such that it can be cited independently in the future. For instructions see: http://journals.plos.org/plosone/s/submission-guidelines#loc-laboratory-protocols

We look forward to receiving your revised manuscript.

Kind regards,

Evangelia V. Avramidou, PhD

Academic Editor

PLOS ONE

Journal Requirements:

2. In your Methods, please specify the exact source(s) of the cassava and potato extracts used in your study.

Additional Editor Comments:

Dear Dr. Yaxin Sang,

Reviewers' comments:

Reviewer's Responses to Questions

**Comments to the Author**

1. Is the manuscript technically sound, and do the data support the conclusions?

Reviewer #1: Yes

Reviewer #2: Yes

2. Has the statistical analysis been performed appropriately and rigorously? 

Reviewer #1: Yes

Reviewer #2: I Don't Know

3. Have the authors made all data underlying the findings in their manuscript fully available?

Reviewer #1: No

Reviewer #2: No

4. Is the manuscript presented in an intelligible fashion and written in standard English?

Reviewer #1: Yes

Reviewer #2: Yes

5. Review Comments to the Author

Reviewer #1: Overview: The manuscript presents adequate data to support use of droplet digital PCR to assess the accurate composition of cassava starch when mixed or blended with starch from other sources such as sweet potato, potato and corn. Are there some industry quality standards required for such starch products be of a certain proportion like there is for varietal wines? Is there any reference that can provide documentation of fraud by food processor? Are there health concerns (e.g. associated food allergies)? Authors indicate models were developed but the data only shows target DNA copies and correlations only. Does this count as a model? Otherwise, the science and methodology appear solid.

Specific comments:

Ln 29-30. Cassava DNA- clarify the linear relationship was with regard cassava dry weight to DNA concentration.

Ln 40-43. Recommend rewriting the first five sentences with a reference to cassava adulteration by other starch produces due to economics.

Ln 45. Sanitation tests?

Ln 53. Digital dPCR (ddPCR) and use ddPCR thereafter in this manuscript.

Ln 184-190. Needs a Table or a figure.

Ln 196 – Fig. 5 is missing.

Table 5 is incoherent without adding what the samples are. The results indicate cassava/sweet potato at different ratios which need to be included in the title or at least foot-noted.

Ln 199 maximum relative error 10.2%. Is this shown in Table 5?

Reviewer #2: The manuscript entitled ‘A quantitative study of the adulteration of cassava components in starch products by droplet digital PCR’ describes the development of a method aiming to enable the rapid and accurate detection and quantitation of cassava adulteration in starch products by utilizing droplet digital PCR technology (dd PCR).

The experiment seems well thought out and carefully executed. However certain points need to be elaborated and explained further.

1. It is not clear what was the basis of selecting the Acc gene for the PCR analysis. In fact there is no information or reference given about this gene. Could more genes be studied?

2. Have there been other studies to detect cassava adulteration previously? Have there been studies employing the other PCR technologies and what were the results? Could a comparison between other PCR technologies and dd PCR be provided?

3. It is not clear how a calculation formula was established between cassava quality, DNA concentration and copy number

On line 112 and 113 it is mentioned: “The program for the reaction denaturation was as follows: 95°C, 10 min; 94°C denaturation, 1 min; 56°C annealing, 45 s; 40 cycles; 98°C, 10 min; and 4°C for temporary storage”

The step ‘98°C, 10 min;’ refers to which part of the pcr reaction? Why it is 98°C degrees?

4. On line 127: “The experimental system and operating procedures are shown above” It is not clear what it is shown above.

5. On line 141: On line 127: “The experimental system and operating procedures are shown above” It is not clear what it is shown above.

6. Fig 1: The legend of Fig 1 needs more detail

7. Fig. 2: The legend of Fig 2 needs more detailed description

8. Fig. 3: The same. Please provide some minimal explanation of what the figure and graphs depict.

9. Fig. 5: Fig 5 is missing!

10. The Discussion section is really limited.

Overall the manuscript describes a good effort towards the employment of a rapid and accurate method to detect and quantify cassava adulteration in different starch sources using an important new technology, ddPCR, and merits to be published for reasons of knowledge transfer to the scientific community. However, the introduction, results and discussion sessions need to be described in a much more detailed and elaborate manner providing the reader with much more information on the topic and on data acquisition, in order to make it a comprehensible publication.

6. PLOS authors have the option to publish the peer review history of their article (what does this mean?). If published, this will include your full peer review and any attached files.

Reviewer #1: Yes: Raymond Yokomi

Reviewer #2: No

---

## [Author Response · Author response to Decision Letter 0]

25 Sep 2019

3. Have the authors made all data underlying the findings in their manuscript fully available?

Response: Raw data has been placed in the support file.

Reviewer #1: Overview: The manuscript presents adequate data to support use of droplet digital PCR to assess the accurate composition of cassava starch when mixed or blended with starch from other sources such as sweet potato, potato and corn. Are there some industry quality standards required for such starch products be of a certain proportion like there is for varietal wines? Is there any reference that can provide documentation of fraud by food processor? Are there health concerns (e.g. associated food allergies)? Authors indicate models were developed but the data only shows target DNA copies and correlations only. Does this count as a model? Otherwise, the science and methodology appear solid.

Response: 

1. There is a national standard for starch products in China (China National Standard for Starch Products GB 2713-2015), and we have cited this standard in the revised manuscript. 

2. Starch adulteration has been reported before and we have cited the reference. 

3. Food allergies are another concern of starch adulteration, and we have added this concern to the introduction. 

4. We used a mixture of sweet potato and cassava starch to model starch adulteration.

---

## [Decision Letter · Decision Letter 1]

23 Oct 2019

PONE-D-19-17769R1

A quantitative study of the adulteration of cassava components in starch products by droplet digital PCR

PLOS ONE

Dear Dr Sang,

Thank you for submitting your manuscript to PLOS ONE. After careful consideration, we feel that it has merit but does not fully meet PLOS ONE’s publication criteria as it currently stands. Therefore, we invite you to submit a revised version of the manuscript that addresses the points raised during the review process.

We would appreciate receiving your revised manuscript by Dec 07 2019 11:59PM. To enhance the reproducibility of your results, we recommend that if applicable you deposit your laboratory protocols in protocols.io, where a protocol can be assigned its own identifier (DOI) such that it can be cited independently in the future. For instructions see: http://journals.plos.org/plosone/s/submission-guidelines#loc-laboratory-protocols

We look forward to receiving your revised manuscript.

Kind regards,

Evangelia V. Avramidou, PhD

Academic Editor

PLOS ONE

Additional Editor Comments (if provided):

Dear authors,

according to reviewers opinion please proceed according to minor revision comments.

Reviewers' comments:

Reviewer's Responses to Questions

**Comments to the Author**

1. If the authors have adequately addressed your comments raised in a previous round of review and you feel that this manuscript is now acceptable for publication, you may indicate that here to bypass the “Comments to the Author” section, enter your conflict of interest statement in the “Confidential to Editor” section, and submit your "Accept" recommendation.

Reviewer #1: All comments have been addressed

2. Is the manuscript technically sound, and do the data support the conclusions?

Reviewer #1: Yes

3. Has the statistical analysis been performed appropriately and rigorously? 

Reviewer #1: Yes

4. Have the authors made all data underlying the findings in their manuscript fully available?

Reviewer #1: Yes

5. Is the manuscript presented in an intelligible fashion and written in standard English?

Reviewer #1: Yes

6. Review Comments to the Author

Reviewer #1: Please see pdf attachment. New edits are suggested to improve clarity of the manuscript. Mass is confusing to this reviewer in the context of this manuscript. It is preferable to use dry weight.

7. PLOS authors have the option to publish the peer review history of their article (what does this mean?). If published, this will include your full peer review and any attached files.

Reviewer #1: Yes: Raymond K. Yokomi

---

## [Author Response · Author response to Decision Letter 1]

20 Nov 2019

Dear editor：

I have completely revised the revised comments and submitted a modified version. Contact if you have any questions.

---

## [Editor Report · Decision Letter 2]

21 Nov 2019

PONE-D-19-17769R2

Identification and quantification of cassava starch adulteration in different food starches by droplet digital PCR

PLOS ONE

Dear Dr Sang,

Thank you for submitting your manuscript to PLOS ONE. After careful consideration, we feel that it has merit but does not fully meet PLOS ONE’s publication criteria as it currently stands. Therefore, we invite you to submit a revised version of the manuscript that addresses the points raised during the review process.

We would appreciate receiving your revised manuscript by Jan 05 2020 11:59PM. To enhance the reproducibility of your results, we recommend that if applicable you deposit your laboratory protocols in protocols.io, where a protocol can be assigned its own identifier (DOI) such that it can be cited independently in the future. For instructions see: http://journals.plos.org/plosone/s/submission-guidelines#loc-laboratory-protocols

We look forward to receiving your revised manuscript.

Kind regards,

Evangelia V. Avramidou, PhD

Academic Editor

PLOS ONE

Additional Editor Comments (if provided):

Dear authors,

I have checked that you adopted reviewers's comments, but I still have some corrections regarding the manuscript. Please fullfill in line 31 do you mean products by "oducts"??

Furthermore, according also to reviewer's and mine opinion you should substitute word "mass" with "weight" also to tables and figures. Please make tha above corrections in order that your article can be published.

With kind regards

---

## [Author Response · Author response to Decision Letter 2]

17 Jan 2020

Reply

A rebuttal letter

Ln 2-3. Article title has been changed

Ln 7. Text has been changed

Ln 8 The text was changed to weight

Ln 12 Text has been changed

Ln 13 Text has been changed

Ln 22 Text has been changed

Ln 24. Text has been changed

Ln 26 Text has been changed

Ln 30-31. Text has been changed

Ln 34 Text has been changed

Ln 36 Text has been changed

Ln 49. Text has been changed

Ln 51 Text has been changed

Ln 57 Text has been changed

Ln 68. Text has been changed

Ln 85 Text has been changed

Ln 88 Text has been changed

Ln 110 Text has been changed

Ln 117. Text has been changed

Ln 148. Text has been changed

Ln 154 Text has been changed

Ln 161 Text has been changed

Ln 171-175 Rewritten

Ln 224 Text has been changed

Ln 227 Text has been changed

Ln 297 Text has been changed

Ln 299 Text has been changed

Ln 301 Text has been changed

Ln 303-311. Text has been changed

---

## [Editor Report · Decision Letter 3]

22 Jan 2020

Identification and quantification of cassava starch adulteration in different food starches by droplet digital PCR

PONE-D-19-17769R3

Dear Dr. Sang,

We are pleased to inform you that your manuscript has been judged scientifically suitable for publication and will be formally accepted for publication once it complies with all outstanding technical requirements.

With kind regards,

Evangelia V. Avramidou, PhD

Academic Editor

PLOS ONE
---

## [Editor Report · Acceptance letter]

28 Jan 2020

PONE-D-19-17769R3 

Identification and quantification of cassava starch adulteration in different food starches by droplet digital PCR 

Dear Dr. Sang:

I am pleased to inform you that your manuscript has been deemed suitable for publication in PLOS ONE. Congratulations! Your manuscript is now with our production department. 

With kind regards,

on behalf of

Dr. Evangelia V. Avramidou 

Academic Editor

PLOS ONE